Clusters explaining the relation between menopause and self-reported periodontal disease: a cross-sectional study

http://orcid.org/0000-0001-9113-2399 Fadel Hani T. 1 hani.fadel@yahoo.com
Qarah Lujain A. 2
Alharbi Manal O. 3
Al-Sharif Alla 1
Al-Harkan Doaa S. 4
Kassim Saba 1
http://orcid.org/0000-0002-3416-165X Abu-Hammad Osama 5
Dar-Odeh Najla 5
1 Department of Preventive Dental Sciences, College of Dentistry, Taibah University , AlMadinah AlMunawwarah , Saudi Arabia
2 Ministry of Health , AlMadinah AlMunawwarah , Saudi Arabia
3 Islamic University in Madinah , AlMadinah AlMunawwarah , Saudi Arabia
4 Department of Oral & Maxillofacial Diagnostic Sciences, College of Dentistry, Taibah University , AlMadinah AlMunawwarah , Saudi Arabia
5 School of Dentistry, University of Jordan , Amman , Jordan
Abu Hasna Amjad
Electronic publication date: 2025 Jan 27
Publication date: 2025
Volume: 13
Electronic Location ID: e18861
Received 2024 Oct 9; Accepted 2024 Dec 21
Copyright: © 2025 Fadel et al.
Copyright year: 2025
Copyright holder: Fadel et al.
License: This is an open access article distributed under the terms of the Creative Commons Attribution License, which permits unrestricted use, distribution, reproduction and adaptation in any medium and for any purpose provided that it is properly attributed. For attribution, the original author(s), title, publication source (PeerJ) and either DOI or URL of the article must be cited.
License URL: https://creativecommons.org/licenses/by/4.0/

Keywords: Dry mouth, Menopause, Oral health, Periodontal disease, Self-report, Women

Funding: The authors received no funding for this work.

==============================
Background

Menopause is an important milestone in the women’s life continuum and is associated with potentially adverse effects, including those related to oral health. This study assessed self-reported periodontal disease in relation to menopausal status.

Methods

A cross-sectional study involving a convenience sample of female university dental hospital attendees was conducted using a validated, self-administered, self-reported periodontal disease questionnaire. A two-step cluster analysis was used to categorize the participants based on menstrual period (MP) continuity, systemic diseases and age. Differences between clusters were analyzed using chi-square test.

Results

From 112 included participants, three clusters resulted from the analysis: Cluster #1 (37 ± 8 years, no systemic diseases and continued MP), Cluster #2 (40 ± 10, with systemic diseases and continued MP) and Cluster #3 (54 ± 9, with systemic diseases and discontinued MP). Cluster #3 tended to have less optimal oral hygiene habits and more missing teeth (p > 0.05). Clusters #1 and #2 insignificantly reported more gingival bleeding, tooth sensitivity and calculus (p > 0.05). Cluster #3, on the other hand, presented with more self-reported oral dryness (p ≤ 0.05).

Conclusions

Within study limits, clusters of menopausal women with systemic diseases reported high symptoms of periodontal disease that were not significantly different from younger individuals, with the exception of oral dryness.

Introduction

Menopause is defined by the World Health Organization (WHO) as “the permanent cessation of menstruation due to loss of ovarian follicular activity” (World Health Organization, 1996). It is diagnosed retrospectively following 12 months of amenorrhea that is not associated with pathology. The term “Menopause” itself means “without estrogen”, and refers to the time at which cyclic ovarian function—known as menstruation—ceases (World Health Organization, 1996). As levels of circulating sex hormones change, several clinical outcomes ensue with a resulting potential impact on emotional state and quality of life.

Within the context of oral health, Mariotti (2005) suggested that hormonal fluctuations during the different stages of menopause may lead to certain inflammatory responses in the body. Notably, these fluctuations and changes increase the sensitivity of the gingiva to microbial dental plaque and calculus, consequently increasing the susceptibility to adverse oral health outcomes such as gingivitis, periodontitis, oral dysesthesia and xerostomia or dry mouth (Mosley, Smith & Dezan, 2015; American Academy of Pediatric Dentistry, 2016). Further, Alves et al. (2015) reported that the gingival epithelium becomes thinner, atrophic and more prone to inflammatory changes during menopause. Hormones have been shown to directly and indirectly exert effects on the periodontium. Estrogen and progesterone affect cellular proliferation, differentiation and growth in target tissues, including keratinocytes and fibroblasts in the gingiva (Mariotti & Mawhinney, 2013; Jafri et al., 2015). Researchers therefore propose that the sudden decrease in estrogen production may be associated with primary osteoporosis, which has an effect on the jaw bones (Alves et al., 2015). It has also been suggested that the reduction in bone mineral density resulting from osteoporosis contributes to periodontal disease progression in menopausal women (Alves et al., 2015). Moreover, estrogen may interfere with other periodontal tissues such as the gingiva and periodontal ligament, and influence host immune-inflammatory responses (Alves et al., 2015). Jonasson & Rythén (2016) reported a link between osteoporosis of the mandible and the peripheral skeleton with alveolar bone loss of the mandible and tooth loss in menopausal women.

Interestingly, periodontal disease is less prevalent in women than men. However, Gupta et al. (2018) demonstrated that the opposite takes place in older populations. The currently increasing life span of populations indicates that the proportion of menopausal women affected by periodontal disease is on the rise. The group lead by Gil-Montoya et al. (2021) reported that approximately one in two perimenopausal women is affected by moderate to severe periodontitis, highlighting the need for manual intervention to reduce periodontitis risk.

Despite the available evidence, there are no data that have focused on the personal input of women in Northwest Saudia Arabia with regards to their oral health, namely periodontal disease and the possible relation with their menopausal condition. This can help oral health promotion, early treatment and prevention of further complications associated with different stages of menopause. Specifically, this will aid in the establishment of a referral system between gynecologists, general practitioners and dentists that involves regular dental checkups for early detection and intervention, and based on the fact that oral health is an integral component of the overall health.

The objective of this study was thus to assess the self-reported periodontal disease among a selected group of women attending a university outpatient dental clinic in relation to their menopausal status.

Materials and Methods

Study design

This was a cross-sectional analytical study, conducted at the Taibah University College of Dentistry outpatient clinics in AlMadinah AlMunawwarah, Saudi Arabia. Approval from the Taibah University College of Dentistry Research Ethics Committee was obtained prior to commencement of the study, as all set procedural institutional guidelines were followed (Approval no. TUCDREC/20180107/Fadl).

Ethical considerations

The study was conducted in accordance with the ethical guidelines set by the Declaration of Helsinki (World Medical Association, 2013). Before recruitment, the purpose of the study was explained to each of the participants and informed consent was obtained. They were also informed that participation was voluntary, with no negative repercussions in terms of quality of offered healthcare due to declining to participate. Participants were also assured that all information will be kept confidential and will only be used for research and educational purposes. In addition, participants were informed whenever necessary of their treatment needs based on the findings of this study.

Study population and sample size calculation

The study involved a convenience/consecutive sample from all female individuals aged between 22 and 75 years, who were attending the college clinics between November 2018 and July 2019. Children and individuals below 22 years were excluded alongside those with dementia or mental health or radiotherapy. Approximately 13,000 patients (women and men) visit the clinics each year.

Based on the reported prevalence of the outcome “gingivitis” in the general population of 75%, with a desired level of confidence of 95%, and a margin of error of 10, and an estimated population of women attending the clinics annually of 6,500, a sample size of 97 participants was required for the study (Sample Size Calculation for Cross-sectional Studies with Percentage as Outcome; Chisquares Inc., Sandy Springs, GA, USA). Incomplete responses were to be excluded from the analysis.

Interview and questionnaire

Data were collected by means of an interview and a self-administered questionnaire. The interview was conducted in the waiting area and involved explanation of the study, giving consent to participate and filling the questionnaire. The questionnaire took 3–5 min to complete and included providing information about age, education level, marital status, siblings, current health status and social and oral hygiene habits. Participants were also asked about continuation of their menstrual period (MP). Moreover, a valid and reliable Arabic version of the self-reported periodontal health questionnaire was included (Khader, Alhabashneh & Alhersh, 2014). This section consisted of seventeen close ended dichotomous questions, representing the outcome variables.

Data analysis

Appropriate descriptive and inferential statistics were used. A two-step cluster analysis was performed to identify hidden patterns and relationships by categorizing participants based on the reported background, demographic and health data. Repeated attempts utilizing different demographic and habitual variables produced a clustering model of “Good” quality via Silhouette’s measure. The finally selected clustering predictors, in order of importance; were menstrual period continuation, presence of systemic diseases and age. A post hoc power calculation was performed considering the clusters with the highest (n = 65) and lowest (n = 22) number of participants and a confidence interval of 95% yielded a study power of 77% (OpenEpi, Version 3, open source calculator—PowerCross). A p-value ≤ 0.05 was considered statistically significant. A statistical package was used (version 20; SPSS Inc., Chicago, IL, USA).

Results

Of a total of 114 participating female patients, two were excluded due to incomplete data. The remaining 112 returned their completed questionnaires and agreed to participate in the study while awaiting their treatment appointment at the dental clinics. The mean age of the total sample was 40 (±10) years. Fifty-two percent held a university degree and 66% were employed. Thirty-six percent of the participants suffered from different systemic conditions such as hypertension and diabetes mellitus. Similarly, 59% were on non-specified chronic medication, five of which specifically reporting the use of hormone replacement therapy (HRT). Fourteen percent of the total participants were current smokers, three of them smoked cigarettes and the remaining used waterpipes. Twenty percent had already discontinued their menstrual period.

The two-step cluster analysis resulted in three clusters, with cluster #1 (n = 65) including women with no systemic conditions and cluster #3 (n = 22) involving those with a completely discontinued menstrual period (Fig. 1). Compared to the other two clusters, women in cluster #3 mostly completed up to high school education (p < 0.001), 55% were unemployed (p < 0.05) and had the highest number of children (p < 0.01) (Table 1). With regards to participants from cluster #2, on the other hand, 84% were married (p > 0.05) and 80% were on medication (p < 0.01) (Table 1).

Figure 1 Characteristics of the three clusters resulting from the 2-step cluster analysis, with the predictors “Menstrual Period Continuation”, “Presence of Systemic Diseases” and “Age”.

Table 1 Demographic characteristics of the total sample (n = 112) and the three clusters.

Variable	Total sample (n = 112)	Cluster #1 (n = 65)	Cluster #2 (n = 25)	Cluster #3 (n = 22)	p value	
Marital status—n (%)					0.502	
Not married	27 (24)	18 (28)	4 (16)	5 (23)	
Married	85 (76)	47 (72)	21 (84)	17 (77)	
Education—n (%)					0.000	
High school or lower	54 (48)	26 (40)	12 (48)	20 (91)	
University degree	58 (52)	39 (60)	13 (52)	2 (9)	
Employment—n (%)					0.021	
No	38 (34)	22 (34)	4 (16)	12 (55)	
Yes	74 (66)	43 (66)	21 (84)	10 (45)	
Siblings—n (%)					0.001	
None	27 (24)	17 (26)	4 (16)	6 (27)	
1–2	22 (20)	17 (26)	4 (16)	1 (5)	
3–4	33 (29)	19 (29)	12 (48)	2 (9)	
>4	30 (27)	12 (19)	5 (20)	13 (59)	
Recent medication—n (%)*					0.005	
No	46 (41)	35 (54)	5 (20)	6 (27)	
Yes	66 (59)	30 (46)	20 (80)	16 (73)	
Smokin—n (%)†					0.255	
No	96 (86)	53 (82)	22 (88)	21 (96)	
Yes	16 (14)	12 (18)	3 (12)	1 (4)	
Notes:

p-values in bold fonts are statistically significant using chi-square test at 0.05.

* Only 5 (5%) of those on medication are on HRT.

† Only 3 (3%) smoke cigarettes, the remaining smokers use water pipe.

Cluster #3 apparently had the least percentage of participants who regularly brushed their teeth (91%), used other tooth cleaning aids (41%) or used mouthwash (41%) compared to the other two clusters, although the differences were not statistically significant (p > 0.05) (Fig. 2). The same cluster also tended to have the highest prevalence of tooth loss (96%) and replacement of missing teeth (50%) compared to the other clusters (Fig. 3). However, the differences were insignificant (p > 0.05). With regards to the self-reported periodontal disease (SRPD) questions, the observations of swollen and painful gums, pus, movable teeth, teeth that have changed in position or longer teeth were more reported among individuals in cluster #3, albeit the lack of statistical significance (p > 0.05) (Table 2). The feeling of dry mouth was also significantly higher in cluster #3 compared to the other two clusters (p < 0.05) (Table 2). On the other hand, self-reported bleeding gums, tooth sensitivity and calculus deposits were relatively, yet insignificantly; higher in clusters #1 and #2 (p > 0.05) (Table 2).

Figure 2 A bar graph showing the oral hygiene habits in the total sample (n = 112) and the three clusters.

No significant differences were observed using chi-square test.

Figure 3 A bar graph showing the prevalence of self-reported tooth loss and tooth replacement in the total sample (n = 112) and the three clusters.

No significant differences were observed using chi-square test.

Table 2 Positive answers (YES) to the self-reported periodontal disease questionnaire by the total sample (n = 112) and the three clusters.

Variable	Total sample (n = 112)	Cluster #1
(n = 65)	Cluster #2
(n = 25)	Cluster #3
(n = 22)	p value	
Did you notice any swollen or reddish area in your gums (more than the usual)?	30 (27)	17 (26)	6 (24)	7 (32)	0.820	
Do your gums bleed easily?	50 (45)	30 (46)	12 (48)	8 (36)	0.676	
Do you feel pain from your gums?	28 (25)	12 (18)	8 (32)	8 (36)	0.161	
Does your mouth feel dry?	29 (26)	11 (17)	8 (32)	10 (45)	0.022	
Have you noticed an unpleasant smell from your mouth?	35 (31)	24 (37)	5 (20)	6 (27)	0.271	
Do you have any abscess or pus in your mouth?	13 (12)	7 (11)	1 (4)	5 (23)	0.128	
Does food get trapped between your teeth?	85 (76)	51 (79)	18 (72)	16 (73)	0.755	
Do you feel that your teeth move/have moved?	15 (13)	5 (8)	4 (16)	6 (27)	0.060	
Have you noticed that your teeth have changed in their position?	23 (20)	12 (18)	5 (20)	6 (27)	0.674	
Have you noticed that your teeth have become longer?	18 (16)	8 (12)	4 (16)	6 (27)	0.255	
Do you feel any sensitivity in your teeth?	46 (41)	28 (43)	13 (52)	5 (23)	0.111	
Do you have any calculus or limestone deposits on your teeth?	60 (54)	33 (51)	16 (64)	11 (50)	0.494	
Do you think you have any disease in your gums or tissues that support your teeth (periodontium) or loss of bone that fixes the teeth?	18 (16)	9 (14)	4 (16)	5 (23)	0.618	
Has any dentist or hygienist ever told you that you have deep periodontal pockets?	6 (5)	3 (5)	2 (8)	1 (4)	0.801	
Have you ever been told that you needed treatment for your gums/periodontal tissues?	16 (14)	7 (11)	3 (12)	6 (27)	0.150	
Have you ever received any treatment for your gums/periodontal tissues?	26 (23)	13 (20)	6 (24)	7 (32)	0.522	
Have you ever received any surgical treatment for your gums/periodontal tissues?	3 (3)	3 (5)	0 (0)	0 (0)	0.328	
Note:

p-values in bold fonts are statistically significant using chi-square test at 0.05.

Discussion

This study aimed at evaluating self-reported oral health in women in relation to their menopausal status. Women who have not reached menopausal age insignificantly reported more bleeding gums, tooth sensitivity and calculus deposits than their menopausal counterparts. This is in line with findings from larger registries, reporting more oral health problems by younger women (Azofeifa et al., 2014). An explanation could be that younger women are exposed to continuous hormonal changes, with the potential of exaggerated responses of the dental and periodontal tissues to local stimuli (Dar-Odeh et al., 2017). Furthermore, the fact that women in the reproductive age may go through several pregnancies, are minimally aware at the time with regards to their oral health and often avoid dental visits could all contribute to the characteristic adverse oral health findings (Dar-Odeh et al., 2018).

There was an insignificant tendency for older menopausal women with different systemic comorbidities, i.e., Cluster #3, to observe more painful gums, pus, longer teeth, teeth that are movable, or teeth that have changed their position, all of which are known symptoms related to advanced periodontitis. Recent cohort studies showed that postmenopausal women were more likely to have periodontitis (Park et al., 2023a), and that women with periodontitis are more likely to develop osteoporosis (Choi et al., 2017). Furthermore, it was concluded from previous studies that diabetes mellitus and hypertension are particularly associated with missing teeth among women (Dar-Odeh et al., 2019). A recent nested case-control study showed that healthy patients with periodontitis had a higher mean systolic and diastolic blood pressure than periodontitis-free individuals (Muñoz Aguilera et al., 2021). The increased local and systemic inflammatory markers detected in periodontitis patients are associated with vascular changes and endothelial dysfunction. Furthermore, several studies concluded that diabetic patients had a higher prevalence of periodontal disease than healthy individuals, and attributed the development of diabetes to oral infection and periodontal pathogens (Martínez-García & Hernández-Lemus, 2021).

It was important to investigate the potential role of smoking in this study. Smoking is considered a major risk factor in periodontal disease development and progression. Recent research suggests that smoking acts as a facilitator for the colonization of periodontal pathogens, influencing the structure of the subgingival microbial community and worsening treatment outcomes (Jiang et al., 2020). However, no difference in smoking was observed between the three studied clusters.

Additional findings from this study confirmed what was mentioned earlier, which showed a tendency for menopausal women to have higher prevalence of missing teeth and tooth replacement. Interestingly, the current study identified relatively poor oral hygiene practices among menopausal women, which aggravates the risk for dental caries and periodontal disease, both of which known to lead to tooth loss if left untreated, and eventual tooth replacement to restore function and/or appearance. Moreover, menopausal women were of low educational backgrounds, mostly unemployed and with more children. These sociodemographic variables could play a role in predisposing to low awareness to oral hygiene practices and favorable oral health outcomes. Park et al. (2023b) concluded in their nationwide cohort study that good oral hygiene practices are adversely correlated with osteoporotic fractures.

It was observed that clusters with women who continued to have their menstrual period tended to have more self-reported bleeding gums, tooth sensitivity and calculus deposits, albeit insignificant from others. This can be expected since hormonal fluctuations during menopause can lead increased sensitivity of the gums and teeth to stimuli (Mosley, Smith & Dezan, 2015), and calls for more focused self-performed and professional oral healthcare measures.

Menopausal women with systemic comorbidities reported a significantly higher prevalence of dry mouth. Krupa et al. (2023) have recently concluded that there was a significant association between menopausal duration and salivary flow rates. Older age is associated with polypharmacy which could also predispose to dry mouth (Cannon et al., 2023). On the other hand, several medications are known for improving periodontal health. Non-steroidal anti-inflammatory drugs are thought to enhance periodontal treatment outcomes through their anti-inflammatory effects (Ren et al., 2023). Further, hormone replacement therapy (HRT) during menopause is thought to relieve dry mouth symptoms (Wang et al., 2021). Moreover, HRT seems to enhance alveolar bone density and teeth durability (Pizzo et al., 2011). However, as low as only five participants reported the use of HRT in the current study, which statistically limits its influence on the study outcomes.

The relatively small sample size is looked upon as a study limitations, since it may have impacted the extrapolation of the observed findings alongside the convenience sampling that precluded generalizability of the finding. This may dictate the consideration of the current investigation as a pilot study at best. However, the self-reporting of oral health symptoms in menopausal women in this study covers an important aspect, providing unique information that can be utilized for hypothesis generation in future studies. Moreover, the cross-sectional design limits the identification of temporal relationships between menopause and oral health parameters, necessitating the interpretation of findings with caution. Another limitation is the non-disclosure of the medication types consumed by more than half of the participants, which in theory may have included drugs that increase (anti-cancer drugs) or decrease (anti-inflammatory agents) the risk of periodontitis, although minimal conclusive evidence on such associations is available in the literature (Albandar, Susin & Hughes, 2018).

Conclusions

From this investigation, it can be concluded that clusters of menopausal women with systemic diseases reported high symptoms of oral and periodontal disease that were not significantly different from younger individuals, with the exception of oral dryness. Further studies with larger study samples are required to expand on such findings.

Supplemental Information

Supplemental Information 1 Study Data.

Supplemental Information 2 STROBE Checklist.

Additional Information and Declarations

Competing Interests

The authors declare that they have no competing interests.

Author Contributions

Hani T. Fadel conceived and designed the experiments, analyzed the data, prepared figures and/or tables, authored or reviewed drafts of the article, and approved the final draft.

Lujain A. Qarah conceived and designed the experiments, performed the experiments, prepared figures and/or tables, authored or reviewed drafts of the article, and approved the final draft.

Manal O. Alharbi conceived and designed the experiments, performed the experiments, prepared figures and/or tables, authored or reviewed drafts of the article, and approved the final draft.

Alla Al-Sharif analyzed the data, authored or reviewed drafts of the article, and approved the final draft.

Doaa S. Al-Harkan analyzed the data, authored or reviewed drafts of the article, and approved the final draft.

Saba Kassim analyzed the data, authored or reviewed drafts of the article, and approved the final draft.

Osama Abu-Hammad conceived and designed the experiments, analyzed the data, authored or reviewed drafts of the article, and approved the final draft.

Najla Dar-Odeh conceived and designed the experiments, authored or reviewed drafts of the article, and approved the final draft.

Human Ethics

The following information was supplied relating to ethical approvals (i.e., approving body and any reference numbers):

Taibah University Research Ethics Committee.

Data Availability

The following information was supplied regarding data availability:

The raw data presented as participant responses to questions in one single file.

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
