# Peer review of "Clusters explaining the relation between menopause and self-reported periodontal disease: a cross-sectional study"

_PeerJ, doi:10.7717/peerj.18861_

## Round 0.1 · original submission · Minor Revisions

Dear authors,

Thank you for your submission. The manuscript is of high relevance but would benefit from several minor revisions. In the Introduction, it is recommended to add expert opinions supporting the research. In the Materials and Methods section, the Data Analysis section could be more concise, focusing on the analyses used. Updating references with more recent sources (post-2020) would strengthen the manuscript. Additional points include clarifying the effect of smoking on gingival condition, the rationale for including smokers, addressing potential effects of medications (especially NSAIDs and HRT) on inflammation and periodontal health, and explaining the impact of participant distribution across clusters.

Reviewer 1 has also provided an annotated PDF

·

Basic reporting

No comment

Experimental design

No comment

Validity of the findings

No comment

Additional comments

Please revise the manuscript, add the discussion with systemic disease hypertension and diabetic with periodontal disease based on result.

Reviewer 2 ·

Basic reporting

Abstract:
The result section: What is the reason for choosing a subject age under 40 years old when what is being discussed is menopause (which generally occurs at an age above 45 years old)
Introduction:
It is recommended that some expert opinions be included that support this research.
The material and methods section:
Marital stats or marital status?
Data Analysis:
It is recommended to be shorter and focus on explaining the analysis used.
Conclusion:
It would be better to remove the words "within limitations of this study" in the conclusion

Experimental design

In Figure 2, it is suggested that insignificant results are not displayed.

Validity of the findings

No comment

Additional comments

The research question well defined, relevant & meaningful. It is stated how research fills an identified knowledge gap

·

Basic reporting

References are related but a bit outdated most references are before 2020,,, more references after 2020 would be recommended

Experimental design

No comment

Validity of the findings

No comments

Additional comments

-It was not clear whether smoking had any effect on the gingival condition knowing that 16 out of 112 participants were smokers.

- Why weren't the smokers excluded from the study?

- Cluster # 1 included more participants (n= 65) than cluster # 2 (25) and cluster # (22) how did this affect the study results?

- 66 participants were on recent medication 5% of which were on HRT, how about the rest any medication that can affect the degree of inflammation (NSAID/.....)?
knowing that in cluster # 2 (20 out of 25 were under medication) and 16 out of 22 in cluster # 3 were also under medication.
what kind of medication? did they have any effect on the gingival and periodontal tissues ?

---

## Round 0.2 · Minor Revisions

Dear authors,

The reviewers have raised additional minor concerns. Kindly address their comments thoroughly.

·

Basic reporting

Clear

Experimental design

Clear within Aims and Scope of the journal

Validity of the findings

Clear

Additional comments

Please checked the new comment in file attached

Reviewer 2 ·

Basic reporting

The abstract reIn the abstract, a summary of the flow of the paper's contents is clearly described.port overview has been written clearly.

In the introduction, it is better to inform the opinions of experts who support direct research in sentences.

In materials and methods section: have been clearly described

Data analysis: The analysis data still uses long sentences; it is suggested that they be shortened to be more explicit.

Conclusion: has been explained clearly

Experimental design

Tables: Looks clearer just have to match the format

Validity of the findings

No comment

Additional comments

The research question is well defined, relevant & meaningful.

---

## Round 0.3 · accepted · Accept

Dear authros,

We are pleased to inform you that your manuscript has been accepted for publication following a thorough review process. The reviewers and editorial team commend the clarity, scientific rigor, and originality of your work, which makes a significant contribution to the field.

·

Basic reporting

clear

Experimental design

Original primary research within aims and scope of the journal

Validity of the findings

Coclusions are well stated

Additional comments

The author has revised the masuscript

Reviewer 2 ·

Basic reporting

revision done

Experimental design

revision done

Validity of the findings

revision done

Additional comments

No

·

Basic reporting

no comments

Experimental design

no comments

Validity of the findings

no comments

Additional comments

thank you for your response